# Enhancing Phosphorus and Nitrogen Uptake in Maize Crops with Food Industry Biosolids and *Azotobacter nigricans*

**DOI:** 10.3390/plants12173052

**Published:** 2023-08-25

**Authors:** Sara-Luz Vera-García, Felipe-Neri Rodríguez-Casasola, Josefina Barrera-Cortés, Arnulfo Albores-Medina, Karla M. Muñoz-Páez, Rosa-Olivia Cañizares-Villanueva, Ma.-Carmen Montes-Horcasitas

**Affiliations:** 1Biotechnology and Bioengineering Department, Center for Research and Advanced Studies of the National Polytechnic Institute, Zacatenco Unit, Mexico City, CP 07360, Mexico; saraluz.vera@cinvestav.mx (S.-L.V.-G.); rcanizar@cinvestav.mx (R.-O.C.-V.); cmontes@cinvestav.mx (M.-C.M.-H.); 2National School of Biological Sciences, Environmental Systems Engineering, Adolfo López Mateos Professional Unit, Zacatenco, Mexico City, CP 07738, Mexico; felipenerirodrig@yahoo.com.mx; 3Toxicology Department, Center for Research and Advanced Studies of the National Polytechnic Institute, Zacatenco Unit, Mexico City, CP 07360, Mexico; aalbores@cinvestav.mx; 4CONACYT—Institute of Engineering, Juriquilla Academic Unit, National Autonomous University of Mexico, Queretaro, CP 76230, Mexico; kmunozp@iingen.unam.mx

**Keywords:** development of plants, plant–bacteria interaction, sewage sludge, sustainable fertilizers

## Abstract

The problem of phosphorus and nitrogen deficiency in agricultural soils has been solved by adding chemical fertilizers. However, their excessive use and their accumulation have only contributed to environmental contamination. Given the high content of nutrients in biosolids collected from a food industry waste treatment plant, their use as fertilizers was investigated in *Zea mays* plants grown in sandy loam soil collected from a semi-desert area. These biosolids contained insoluble phosphorus sources; therefore, given the ability of *Azotobacter nigricans* to solubilize phosphates, this strain was incorporated into the study. In vitro, the suitable conditions for the growth of *Z. mays* plants were determined by using biosolids as a fertilizer and *A. nigricans* as a plant-growth-promoting microorganism; in vitro, the ability of *A. nigricans* to solubilize phosphates, fix nitrogen, and produce indole acetic acid, a phytohormone that promotes root formation, was also evaluated. At the greenhouse stage, the *Z. mays* plants fertilized with biosolids at concentrations of 15 and 20% (*v*/*w*) and inoculated with *A. nigricans* favored the development of bending strength plants, which was observed on the increased stem diameter (>13.5% compared with the negative control and >7.4% compared with the positive control), as well as a better absorption of phosphorus and nitrogen, the concentration of which increased up to 62.8% when compared with that in the control treatments. The interactions between plants and *A. nigricans* were observed via scanning electron microscopy. The application of biosolids and *A. nigricans* in *Z. mays* plants grown in greenhouses presented better development than when *Z. mays* plants were treated with a chemical fertilizer. The enhanced plant growth was attributed to the increase in root surface area.

## 1. Introduction

Nitrogen and phosphorus are two of the most essential nutrients for the proper development of plants [1]. However, the application of inappropriate agricultural practices, the growth of urban sprawl, and the geographical location of the land are factors that affect their availability in agricultural soils [2,3,4]. Adding chemical fertilizers to soils has been a strategy for increasing agricultural production; however, the low assimilation of this type of product has contributed to soil degradation and environmental contamination [5,6]. So, considering the importance of ensuring agricultural production, sustainable fertilizers, such as biosolids, are under study [7].

A biosolid is a residue from municipal or industrial wastewater treatment that can be in the solid, semi-solid, or liquid phase [2]. Nine million tons (Mt) of biosolids are produced annually in Europe, along with 17.8 Mt in the United States and 9.7 Mt in Mexico [3,8]. This type of waste usually contains essential elements, such as nitrogen and phosphorus, making it a viable option as a fertilizer for supplementing the necessary nutrients in the soil. Fertilizers of this type are of interest for maintaining a sustainable agricultural system with economic and environmental benefits [4,9,10,11]. Incorporating this material improves soil’s physiological and chemical properties, thus increasing its productivity [12,13]. However, given the origin of biosolids, various federal entities in each region regulate their use as fertilizers to avoid possible environmental and health risks [14,15,16,17]. In Mexico, the federative entity is the Ministry of Environment and Natural Resources (SEMARNAT); in the United States, this is the Environmental Protection Agency (EPA); and in Europe, it is the EUR-Lex Directive [18,19,20].

It is believed that the sludge from wastewater treatment in the food industry has the property of a low health risk in use as fertilizer in agricultural soils. However, regardless of their origin, biosolids must be characterized prior to their application to guarantee their best use and reduce the risk of environmental impacts [3]. The essential nutrients (nitrogen, phosphorous, sulfur), trace elements, and organic compounds that this type of biosolid can provide improve the physicochemical properties of soils, which allows an increase in the yield of agricultural crops [21,22,23]. Biosolids from the food industry were evaluated by Bıyıklı et al. [21] in the cultivation of corn plants. According to these authors, the application of biosolids in concentrations of 0, 30, 60, and 90 tons per hectare prior to planting had positive effects on the growth of corn plants. Santacoloma et al. [22], who applied biosolids from the food industry to coriander crops, reported higher percentages of seed germination.

The microbial flora associated with plants and biosolids have the property of changing the dynamics of the transport and assimilation of nutrients (phosphorus and nitrogen), which favors their use for plant development [2]. Insoluble nutrients can be assimilated by both arbuscular mycorrhizal fungi (AMF) and *Trichoderma* sp., which solubilizes phosphorus and produces phytohormones, as well as by plant-growth-promoting bacteria (PGPB) that reside in the rhizosphere [24,25]. It has been reported that the application of PGPB in agricultural crops can replace chemical fertilizers considering that they improve soil fertility, increase agricultural production, and contribute to environmental conservation [26,27,28]. PGPB can also protect plants from diseases and pests and increase their resistance to adverse conditions, such as drought and water stress [26]. Such is the case of *Azotobacter* sp., a genus of bacteria known for its ability to fix nitrogen, solubilize phosphorus, and produce phytohormones, such as indole acetic acid (IAA). The inoculation of plants with bacteria of the genus *Azotobacter* sp. has allowed the improvement of the quality of the soil and the production of healthy plants [29,30,31,32].

Maize (*Zea mays*) is one of the most important crops worldwide for both human and animal nutrition, and its production has significantly increased in recent decades [33]. It is estimated that in 2020, around 1974 million tons of corn will be grown in the world, with the United States being the main producer with 36% of world production [33]. Corn is a versatile crop that is used to produce various food and non-food products, such as food, beverages, biofuels, and biodegradable plastics, among others. However, due to population growth and food demand, it is necessary to increase maize production worldwide [34]. Diverse sources of nitrogen and phosphorus have been evaluated to identify those that are the most effective for plant development. Of these (mineral and organic), the organic ones based on compost and biosolids produced in water treatment plants have been reported as the most effective fertilizers due to their prolonged effects [4,35,36]. Regarding phosphorus, both soluble and insoluble sources of phosphorus have been evaluated in combination with rhizobacteria and phosphate-solubilizing bacteria [32,37,38].

The objective of this study was to evaluate the development of *Z. mays* plants grown in semi-arid soil enriched with biosolids collected from a food industry waste treatment plant and inoculated with *Azotobacter nigricans*. Biosolids are known for their high phosphorus and nitrogen content, which gives them fertilizing properties. Regarding the *A. nigricans* strain, it is assumed that its ability to solubilize phosphates and produce growth hormones in plants could favor the development of an extensive root system in maize plants [39,40]. These characteristics would potentially improve nutrient uptake, even in soils with low moisture retention capacity. Then, it is assumed that the joint application of biosolids and the *A. nigricans* strain will favor the development of *Z. mays* plants. The evaluation was carried out in vitro and at the greenhouse level. The in vitro experiments evaluated (1) the effects of biosolids on the growth of *Z. mays* plants, (2) the ability of *A. nigricans* to solubilize phosphates, and (3) the effects of inoculation on the growth of *Z. mays* plants. Plant–bacteria interactions were observed via scanning electron microscopy. The results of the in vitro study were applied to the growth of *Z. mays* plants in a greenhouse.

## 2. Results

### 2.1. Physicochemical Properties of Soil and Biosolids

The physicochemical and biochemical parameters obtained from the analysis of the studied solid matrices are presented in Table 1.

According to an analysis conducted by Peck and Soltanpour [41] for agricultural soils, the soil sample selected in this study was sandy loam soil with a slightly basic pH and a slightly saline nature but a high organic matter content. The metal concentration in the soil fell within the recommended values, as indicated in the far-right column of Table 1.

The soil presented a sandy loam texture with a color ranging from light brown to gray (Appendix A) and a high content of sand (71%) and silt (29%). The low fraction of clay made it an easily degradable soil. However, its high silt content was an indicator that it had a high organic matter content (8.7%), which was corroborated by soil samples extracted 5 cm from its surface. The moisture percentage of 27.5% in the soil samples extracted at a depth of 0.3 m was consistent with the low clay content in the soil structure.

The nitrogen and phosphorus content in the soil was determined in the form of nitrate (NO_3_^−^), total nitrogen (N), phosphate (PO_4_^−3^), and total phosphorus (P), with the corresponding values of 11.32 mg kg^−1^, 609.42 mg kg^−1^, 4.44 mg kg^−1^, and 13.62 mg kg^−1^, respectively (Table 1). The concentration of nitrates and total phosphorus were lower when compared to the minimum (30 mg kg^−1^) and maximum (50 mg kg^−1^) values recommended for agricultural soils, respectively [42]. The deficient content of N and P in the form of nitrate and total phosphorus suggested that the addition of nutrients to the soil would be convenient for the better development of agricultural crops.

The physicochemical properties of the biosolids are presented in the third column of Table 1. A pH of 6.8 indicates that there was a neutral biosolid. The organic matter concentration was 0.87%. This value is low compared with that of 1.85% determined by Bıyıklı [21] in biosolids collected from a wastewater treatment plant from the food industry. The nutritional elements phosphorus and nitrogen, in the form of nitrate and phosphate, were determined at concentrations of 320.1 mg kg^−1^ and 27.58 mg kg^−1^, respectively. These values were 28 and 6 times higher than those found in soils.

The concentrations of total solids (TSs), which involved the biomass content, total volatile solids (TVSs), which comprised the amount of organic and inorganic matter, and total organic carbon (TOC) were determined to be at levels of 15.9 g L^−1^, 9.99 g L^−1^, and 631 mg L^−1^, respectively. The chemical oxygen demand (COD) was 6050 mg L^−1^. This value was consistent with what was reported in the literature, which told us that the TOC is 10% of the COD [43]. Of the total microbial count, 1.9 × 10^−5^ ± 5.8 × 10^−9^ CFU mL^−1^ were identified as coliforms (Appendix A). The coliform count was lower than the maximum limit of 2 × 10^−3^ ± 5.2 × 10^−7^ CFU mL^−1^, which was indicated in U.S. EPA PART 503 [19], which established the maximum permissible limit of total coliforms in biosolids.

The presence of heavy metals in biosolids varies depending on the origin and the treatment to which they are subjected. In the present study, Hg, Pb, and Cd were found in a higher concentration than those identified in the soil; however, its concentration was lower than the ceiling concentrations reported in EPA [19].

### 2.2. Activity of Azotobacter Nigricans

#### 2.2.1. Inoculation Time Determination

The growth kinetics of *A. nigricans* in complete Rennie’s medium (CR) and biosolids at 25% (*v*/*v*) are presented in Figure 1. The exponential growth phase began at hour 6 and ended at hour 48. Then, samples of this culture were taken after 24 h to inoculate the Fähraeus systems and other systems that included the study of the effect of this microorganism on plant development. At the time of 24 h, the microbial count was 1 × 10^8^ ± 6.4 × 10^6^ CFU mL^−1^, which was equivalent to 2 ± 0.15 mg mL^−1^.

#### 2.2.2. Nitrogenase Activity

The scarcity of nitrogen during the first 2 h for the *A. nigricans* culture was an indicator of the ability of this strain to fix nitrogen (Figure 2). The maximum nitrogen fixation of 354 nM C_2_H_4_ mL^−1^ h^−1^ was recorded at hour 9; subsequently, the activity decreased from the 15th hour. The growth of *A. nigricans* during nitrogen fixation is shown in the same Figure 2; the biomass reached a value of 620 mg L^−1^ at 72 h.

#### 2.2.3. Phosphorus Solubilization

Figure 3 shows the microbial growth of *A. nigricans* and the ability of this strain to solubilize phosphates due to the variation in the concentration of tricalcium phosphate (TCP). It was observed that the increase in the concentration of TCP increased the microbial count, as well as the concentration of solubilized phosphorus. The highest concentration of solubilized phosphorus was 421 mg L^−1^, and this was equivalent to 82% of the initial value. This was obtained at 48 h with the TCP at a concentration of 0.5 g L^−1^. Such a solubilization percentage demonstrates the ability of *A. nigricans* to metabolize insoluble phosphate and take it up for growth. At a concentration of 1.5 g L^−1^, the TCP was solubilized in a greater quantity (650 mg L^−1^); however, the solubilization percentage was 43%. Of the total phosphorus added as dipotassium phosphate (K_2_HPO_4_), 70% was consumed [36].

The variation in pH while the phosphorus (as TCP) was solubilized by means of *A. nigricans* is shown in Figure 4. In each treatment, the culture of *A. nigricans* started with a pH of approximately 6.4. During the growth of *A. nigricans* with the TCP at 1.5 g L^−1^, the pH dropped to 4.4 at hour 48. At this age of the culture, the greatest amount of soluble phosphorus was obtained.

#### 2.2.4. Indole Acetic Acid Production

Verification of the IAA production due to the growth of *A. nigricans* in Rennie’s medium supplemented with tryptophan at a concentration of 5 mg mL^−1^ was performed via thin-layer chromatography (TLC), as shown in Figure 5. The highest IAA production was recorded between 48 and 120 h with a microbial count of 1.2 × 10^9^ ± 9.6 × 10^7^ CFU mL^−1^ and a pH of 4.8. The calculated Rf values of 0.88 and 0.89 were consistent with those calculated by using the IAA standard, whose Rf value was 0.89. HPLC analysis of the *A. nigricans* culture samples revealed IAA concentrations of 32.62 µg mL^−1^ and 47.44 µg mL^−1^ in culture samples extracted at 48 and 120 h, respectively.

### 2.3. Development of Zea mays Seedlings in Fähraeus System

#### 2.3.1. Seed Germination of *Zea mays*

Of 100 seeds sown in water agar, 95% germinated in 4 days, producing non-contaminated roots with an average length of 5 mm. The seeds were assumed to be viable, considering what was stated by Wang et al. [44]: that seeds with a germination percentage lower than 65% should be discarded.

#### 2.3.2. Effect of Biosolids and *Azotobacter nigricans* on the Development of *Zea mays* Seedlings

The MANOVA of the length of the plants and roots, as well as the stem diameter of the plants due to the effect of time and the addition of biosolids and inoculum in soil, presented significant differences in the cultivation of *Z. mays* plants, as shown in Appendix A. In general terms, the addition of the inoculum favored the growth of the *Z. mays* plants in terms of both stem diameter (F(inoculum) = 216.008; df = 1; *p* < 0.0001; χ^2^ = 0.900) and root length (F(inoculum) = 1077.43; df = 1; *p* < 0.0001; χ^2^ = 0.978). Time, of course, was an important factor (Wilks’Λ = 0.000; F(time)= 1316.319; df = 8; *p* < 0.0001); the plants grew over time, reaching the maximum length after 21 days in all treatments. The maximum length was 35 cm and was obtained in the treatment implemented with a solid matrix prepared with soil, 25% (*v*/*v*) biosolids, and inoculum at a concentration of 1 × 10^8^ ± 6.4 × 10^6^ CFU mL^−1^ (Figure 6). Compared with the positive control (Long Ashton medium) and the soil, both being inoculated with *A. nigricans*, the addition to soil of biosolids by 25% (*v*/*v*) and the inoculum at a concentration of 1 × 10^8^ ± 6.4 × 10^6^ CFU mL^−1^ favored plant growth by 7%. The increases were 11 and 16% higher, respectively, when the inoculum was not added in both systems (soil and Long Ashton medium).

Figure 7 shows the vegetative stage of *Z. mays* on day 9 in the treatments implemented with soil (Figure 7a), biosolids (Figure 7b), 25% (*v*/*v*) biosolids (Figure 7c), and Long Ashton medium (positive control) (Figure 7d). By day 9, the treatments with 25% (*v*/*v*) biosolids (Figure 7c) and the systems inoculated with *A. nigricans* (Figure 7e–h) presented favorable coleoptile and root development compared to those in the other treatments (soil, biosolids and Long Ashton, all without inoculum). In the inoculated treatments, the growth of the coleoptile, the emergence of the first leaf, and the production of lateral roots (thick and thin)—both in the radicle and in the seminal roots—could be clearly appreciated, that is, there was a greater root surface; in these same systems, a coleoptile was recorded as being between 6 and 144% longer and between 7 and 16% thicker than in the treatments without the inoculum. In the treatment that included only biosolids, the development of the coleoptile was between 16 and 24% shorter than that observed in the inoculated treatments (Figure 6); but compared to the positive (Long Ashton medium) and negative (soil) controls, coleoptile length increased by 26–28%. In the treatment implemented with biosolids and *A. nigricans* (Figure 7f), although the roots were as long as in the other treatments, in the treatment with biosolids, they were thicker by 7%. These differences can be better appreciated in the micrographs corresponding to each treatment. 

Figure 8 shows the development of the seedlings on day 21. At this age, the stem was well defined, which allowed the elevation of leaves up to a maximum average length of 35.7 cm in the treatment implemented with seedlings conditioned with 25% *(v*/*v*) of biosolids and inoculated with *A. nigricans*. In the treatments without inoculum (Figure 8a–d), the highest seedling growth reached a height of 32.5 cm, that is, 9.7% shorter than the maximum observed length (Figure 8a). In the treatments without the inoculum (Figure 8a–d), the number of roots decreased by 36–50% compared to the corresponding inoculated treatments (Figure 8e–h). The plants of the soil (Figure 8e) and 25% (*v*/*v*) biosolids (Figure 8g) treatments—both with the inoculum—presented the greatest number of roots (9.5–11), as well as longer roots (42.5 cm).

### 2.4. Colonization of Azotobacter nigricans in the Roots of the Zea mays Plants

The interaction of *A. nigricans* in the roots of the *Z. mays* plants grown in the Fähraeus system with complete Rennie’s medium (CR) and with K_2_HPO_4_, tricalcium phosphate (TCP) (0.5 g L^−1^), and biosolids (25% (*v*/*v*)) as phosphorus sources was observed with a scanning electron microscope (SEM), as shown in Figure 9, Figure 10 and Figure 11, respectively. Figure 9 shows the colonization of *A. nigricans* in a medium with a soluble source of phosphorus. The growth of the microorganism was extensive in the zone of root elongation, as shown in the ×50 SEM image. In addition, the development of the microorganism allowed the growth of root hairs (10 µm). With TCP as a phosphorus source, microbial growth was mainly observed in the root crevices. At a magnification of ×2000, it was notable to observe that the root hairs were not evenly distributed along the root surface but were in the intermediate and upper zones of the tissue of the main root (Figure 10). 

In the treatment implemented with soil and biosolids at 25% (*v*/*v*), the roots presented a greater number of developing root hairs, and it was observed that root colonization occurred in the elongation zone (Figure 11). Colonies of *A. nigricans* were observed to agglomerate on the root surface (SEM × 500). The rhizosphere is the area of the greatest microbial activity and is where a plant absorbs nutrients for its development [45].

### 2.5. Growing Zea mays Plants in a Greenhouse

The growth of *Z. mays* plants due to the effects of the addition of biosolids and fertilizer was analyzed according to both the plant size and the plant dry weight (Figure 12). The ANOVA of the plant size according to the effects of the plant part, inoculum, and biosolids generated significant differences in relation to the plant parts (F(plant part) = 408.667; df = 1; *p* < 0.0001; χ^2^ = 0.791) and to the biosolid concentration (F(biosolids) = 20.390; df = 2; *p* < 0.0001), but the comparison of means by the Tukey’s test for α < 0.05 did not show significant differences. The ANOVA conducted in relation to the plant dry weight (stem and roots) allowed us to demonstrate that the biosolids were a better fertilizer than the fertilizer that was used as a positive control (Figure 12b). The chemical fertilizer was composed of 17% nitrogen (N), 17% phosphorus (P_2_O_5_), and 17% potassium (K_2_O). The ANOVA of the plant dry weight according to the effects of the same variables—plant part, inoculum, and biosolids—generated significant differences, as did that according to the interaction between the inoculum and the biosolids. In this case, the comparison of means by the Tukey test for α < 0.05 showed that the highest concentration of biosolids used, 20%, was the most suitable for growing these plants.

Figure 12a shows the longitudinal growth of plants and roots. The best plant development was obtained in the treatments of soil and biosolids at 20%, both implemented with *A. nigricans*. The average length was of 116.9 cm and 109.8 cm for the plants and 67 cm and 61.8 cm for the roots, respectively.

Compared with the treatments without the inoculum, in general, the treatments implemented with *A. nigricans* presented plants with a greater dry weight (Figure 12B). Figure 13 shows the stem diameter of the plants obtained in the different treatments; of these, the thickest was identified in the treatment implemented with *A. nigricans* and biosolids at 20% (*v*/*w*). On average, the stem diameter was greater than that in the other treatments by 17.5%. As far as the dry weight of the plants is concerned, compared with the positive control (chemical fertilizer), it was higher by 70 and 40% in the treatments corresponding to biosolids at 15 and 20% (*v*/*w*).

The analysis of nitrogen and phosphorus in the treatments with biosolids at 15 and 20% (*w*/*v*) allowed us to observe an increase of 14 and 28% of nitrogen and 54 and 82% of phosphorus, respectively, compared to their respective treatments without inoculation (Figure 14). In the negative control, the increase was 46% of nitrogen and 33% of phosphorus; the positive control did not present an increase in phosphorus. In relation to the negative control (soil), the addition of biosolids increased the nitrogen in the treatments without inoculum by up to 100%. In the inoculated systems (including the control), the increase was smaller, from 49 to 79%. A smaller increase is attributed to the ability of *A. nigricans* to fix nitrogen. Compared to the positive control, biosolids treatments only increased phosphorus assimilation in the inoculated systems; regarding nitrogen, its assimilation increased between 161 and 180%, in the treatments without inoculation, and between 112 and 155% in those inoculated. This was higher in the treatment with the highest concentration of biosolids.

The treatments with 15% (*v*/*w*) and 20% (*v*/*w*) biosolids produced plants with abundant growth of green leaves and without traits of a lack of nutrients, that is, without yellowish coloration or black spots [46]. Plants with the inoculum had longer roots. These plants presented a greater development of the seedlings, and in some, the growth of cobs even began. The seedlings grown with chemical fertilizer presented good size and healthy leaf texture. As in the treatments with biosolids, good root development was observed in the treatment with the inoculum. However, it was not reached until the growth of the cob.

## 3. Discussion

The beneficial use of natural resources and proper soil management requires one to know the physical and chemical properties [47]. Although pH is not a reliable parameter for determining the total acidity of a soil, it is an important property of agricultural soils because it affects plant root growth and the propagation of native soil microflora [48]. Root growth is usually favored in slightly acidic soils with pH values between 5.5 and 6.5. In this work, a pH of 8.1 indicated an alkaline soil in which elements that formed insoluble complexes with phosphorus could affect the immediate availability of this nutrient for both plants and microorganisms [48,49].

Electrical conductivity (EC) is an indirect measure of the number of salts that a soil can contain. According to Corwin and Lesch [50], the value of 1.05 dS m^−1^ describes a very slightly saline soil, that is, soil that favors the absorption of nutrients and water in the plant roots and, consequently, the development of crops.

The contents of sand, clay, and silt give soils their physicochemical properties and determine their hydric properties. It is well known that the sandiest soils are highly permeable, the most clayey ones retain more water, and the siltiest ones favor the retention of water, nutrients, and, consequently, the accumulation of organic matter [51]. Soils with a high silt content are considered fertile [52]. However, the low clay content demands the addition of a greater amount of water for crop growth. The low moisture content of the sandy loam soil studied here was corroborated by soil samples extracted at a depth of 0.3 m.

Organic soil particles come from the products of the microbial decomposition of dead plants, animals, and microorganisms [46]. In the present study, the organic matter content, which was determined through ignition, may have come from organic remains and may have helped to increase the nutrient content of the soil and the cation exchange capacity [40]. These factors make this parameter especially useful for indirectly knowing the fertility of a given soil. The value reported 5 cm from the surface, which indicated that the soil had a large amount of organic matter.

The chemical and nutritional properties of soil depend on the amounts of the different minerals that compose it, such as nitrogen, phosphorus, potassium, calcium, and magnesium, which, although they must be abundant, must also be in balance for the proper development of plants [46]. Usually, the aforementioned minerals are found in soils in the form of insoluble molecular complexes, such as nitrates and phosphates. Nitrates can remain mobile in the aqueous phase of the soil and are susceptible to volatilization and leaching through the different layers of the soil. Phosphates also tend to bind strongly to other elements, which restricts their mobility and bioavailability, thus limiting, as with nitrates for plant growth [53]. Since the exchange capacity of anions in soil is low compared to that of cations, their degradation is necessary for bioavailability in plants [54]. However, given the low concentrations at which both nitrate and phosphate were found in the soil under study, the addition of additional sources of nitrogen and phosphorus is recommended to respond to the requirements of plants for their proper development.

If mineral elements—especially sodium or heavy metals—are present in excess in soil, plant growth can be negatively affected. Some plants can tolerate excess mineral elements, and a few species—for example, halophytes in the case of sodium—grow in these extreme conditions [55]. In the soil used in the present study, the heavy metals identified were within the ranges of the recommended values according to the standard in Title 40, Protection of Environment, part 503 [19]; therefore, the soil’s application is adequate for propagating agricultural crops.

The availability of information on the physicochemical properties of biosolids allows better use thereof. The pH provides information about the content of mineral elements. Organic matter is a measure of the carbon content, which can be used by organisms associated with a plant [48]. In the present study, the value of 0.87% for organic matter was low, but in this case, it was compensated by the concentration of organic matter contained in the agricultural soil under study (8.7%).

The content of organic and inorganic matter, metals, and biomass present in biosolids depends on their origin, as well as the treatments to which they are subjected [56]. In the present study, the components of interest were phosphorus, nitrogen, the microbial count, and various carbon sources, such as biomass, which are essential for the microbial metabolism of the rhizosphere associated with plants, as well as a source of nutrients for plant development.

Heavy metals are of special importance due to their toxic properties, but their presence in biosolids varies depending on the origin and possible treatments to which the biosolids could be subjected [57]. Sludge from the food industry tends to have a low metal content, and its use as a fertilizer is an attractive option. Despite the low content of heavy metals determined in the biosolids under study, their concentrations in plants may increase because of their bioaccumulation through repeated and abundant applications. To avoid this type of situation, it is advisable to conduct periodic monitoring of toxic substances [16].

The nitrogenase activity depends on the species of the strain under study and the carbon source used. Nosrati et al. [58] reported that cultures of *Azotobacter chroococcum* and *Azotobacter vinelandii* exhibited nitrogenase activities of 12.1 and 326.4 nmol C_2_H_4_ h^−1^ vial^−1^ (vial volume = 12 mL; volume of culture medium = 5 mL), respectively, when glucose was used as the carbon source. In the present study, the nitrogenase activity with *A. nigricans* was 352.9 C_2_H_4_ h^−1^ mL^−1^ when using sucrose as a carbon source. This value is 4.4 times higher than that reported by these authors. Nitrogen-fixing bacteria can also metabolize unconventional carbon sources [59,60]. The nitrogenase activity of *A. nigricans* was 125 C_2_H_4_ h^−1^ mL^−1^ when using kerosene as a carbon source [61].

The abilities of *Azotobacter* sp. as a phosphorus solubilizer have been previously reported. Garg et al. [62] evaluated the availability of phosphorus in cow manure (2.27 g L^−1^), obtaining 0.34 mg L^−1^ of soluble phosphate. Phosphate solubilization indices (PSIs) of 3.5 for *A. vinelandii* and 1.1 for *A. chroococcum* were reported by Nosrati et al. [58] in trials carried out with TCP (2.5 g L^−1^) and glucose as a carbon source. Saha-Pal et al. [53] reported the ability of *A. chroococcum* MAL-201 to solubilize 52% of TCP (1 g L^−1^) in culture media prepared with glucose. In the present work, *A. nigricans* solubilized 82% of the TCP added at a concentration of 0.5 g L^−1^ but while using sucrose as a carbon source. Nautiyal [60] reported that the carbon source plays a significant role in the microbial mechanism of solubilizing TCP, and the solubilization level depends on the nature of the compound, as well as on the microorganism involved. The solubilization of TCP has been attributed to the presence of organic acids in culture media [63]. According to Paredes-Mendoza and Espinosa-Victoria [64], organic acids play a key role in the TCP solubilization process.

IAA is a reciprocal signaling molecule in plant–microbe interactions, and it maintains a symbiotic relationship that has evolved between host plants and their microbial allies [65,66,67]. This phytohormone controls almost all aspects of plant growth and development. It plays a fundamental role in cell division, elongation, and root development, and it modifies the architecture of the root system, leaves, and flowers [68]. Most IAA is transported throughout the plant through the phloem, forming concentration gradients and accumulating in different tissues. A well-developed root system is essential for the absorption of water and nutrients and for the anchoring of plants in the soil [69]. The plant reciprocates by providing its bacterial partner with exudates. In the present study, the IAA concentration produced by *A. nigricans* was 47.44 µg mL^−1^ when 5 μg mL^−1^ of tryptophan and sucrose were added as carbon sources. This value was lower than that reported by Das [30], of 9.4 and 16.5 mg L^−1^ per mg of protein from *A. chroococcum* CBD15 and *A. vinelandii* UW, respectively, in culture media prepared with glucose and tryptophan at a concentration of 50 μg mL^−1^ (tryptophan). However, it is important to mention that the carbon source and culture conditions are determining factors in microbial metabolism [67].

Seed germination percentage has been used as an indicator of seed age and vigor (measured in terms of seedling length and root length), as well as an indicator of seedling exposure to pollutants [44,70]. In the present study, certified seeds were used, and the germination study was carried out with the objective of corroborating said certification. Growing conditions also determine the germination level of the seeds [71]. In the present study, the cultivation of seeds in sterilized water agar as a solid support, and since germination was performed with washed seeds grown in an aseptic environment, could also have contributed to the high seed germination percentage obtained (95%). Germination percentages of 80–90% are considered adequate in young and vigorous seeds.

A Fähraeus system is a system that allows the simulation of the development of roots and the rhizosphere in seedlings [72]. In the present study, this system was used to observe the benefits of using biosolids as fertilizer and *A. nigricans* as growth-promoting bacteria in *Z. mays* plants.

For their development, plants require nutrients and other substances that are obtained through the rhizosphere associated with them [73]. In the present study, this role was attributed to *A. nigricans* considering that the seedlings with the highest growth and root development were observed in the treatments implemented with biosolids at a concentration of 25% (*v*/*v*) and inoculated with *A. nigricans* at a concentration of 1 × 10^8^ CFU mL^−1^. The ability of *A. nigricans* to fix nitrogen and solubilize phosphates has already been reported [32,58]. However, in the present work, the phosphates to be solubilized were provided not only by the TCP added in some treatments, but also by the biosolids that were studied. It has been reported that phosphorus deficiency in the roots causes drastic changes in the development of the root system, generally increasing the number of root hairs and lateral roots, which increases the root surface and, therefore, the phosphorous assimilation in plants [74]. Contact of the root tip with a low-phosphorous medium results in the interruption of primary root growth, including reduced cell elongation, cell division, and meristhematic activity. In the number of roots, there were no significant differences, since maize is a monocotyledonous plant that generates lateral roots and, so, it mainly develops in terms of the length and appearance of root hairs.

Even though biosolids are considered a source of nutrients and organic matter [40], the *Z. mays* seedlings presented deficient development when they were added at a 100% concentration; a concentration of 25% (*v*/*v*) was favorable for the development of bending strength plants. It was assumed that the ability of *A. nigricans* to fix nitrogen and solubilize phosphates favored root development. However, the affectation in stem development on day 21 was attributed to a nutrient imbalance. According to López-Bucio et al. [75], the abundance of roots and their characteristics are given by the bioavailability of nutrients. The deficient development of plants due to an imbalance in nutrients was already reported by Canarini et al. [49].

Root hairs play an important role in plant development by improving the absorption of water and nutrients due to the increase in the interface between the soil and the roots [76]. The growth of root hairs has been the subject of many studies, with a particular focus on the function of the cytoskeleton, ion channels, and intracellular ion gradients, as well as their participation in the interaction with rhizosphere microorganisms [77]. However, there are few studies on the interaction of root hairs with microorganisms such as *Azotobacter*. In the present study, the colonization of the root surface of *Z. mays* by *A. nigricans* was located basically in the zone of root elongation, allowing the development of root hairs. The root hairs appeared as tubular cells without divisions; they emerged from the epidermal cells of the main root and showed a filamentous morphology that was perpendicular to the root tissue [78]. Roots’ interaction with *A. nigricans* did not harm the plants; on the contrary, this helped their development and is the reason why its use as a biofertilizer has been considered [30,31,79]. Viscardi et al. [80] found that the colonization of *A. chroococcum* in tomato roots benefited their development under conditions of drought and saline stress, in addition to improving plant growth.

The dry weight of the plants cultivated in the greenhouse for 120 days allowed us to observe the positive effects of inoculation with A. *nigricans* on both the size of the plants and the development of the roots. The homogeneous growth of the plants in the inoculated systems allowed us to assume the rapid adaptation of *A. nigricans* in the different treatments. It is important to mention that the treatments implemented with biosolids allowed the development of plants with abundant and healthy growth, that is, the plants did not present yellowish coloration, black spots, or any other characteristic visual trait of a plant with a nutrient deficiency [46]. It has already been reported that the use of biosolids as fertilizer increases the dry weight of plants [4]. However, in the present study, this effect was only observed in comparison with the fertilizer used, which generated the lowest yields of plant mass. The biosolids had a pH of 6.8 and the soil had a pH of 8.1; this pH difference could affect the immediate availability of nutrients for the development of plants [35].

Another important aspect observed in the cultivation of *Z. mays* at the greenhouse level was the assimilation of phosphorus and nitrogen. Studies with genetically improved maize plants for phosphorus assimilation in the field have resulted in an increase of 22 to 26% [81]. In the present study, the addition of biosolids and the inoculation with *A. nigricans* allowed for an increase of 63% compared to the uninoculated soil and 23% compared to the inoculated soil. The inoculation presented the advantage of an increase in the nutrient assimilation surface, which was favored by the increase in the number and length of roots in the treatments implemented with *A. nigricans*. Increased root length and branching can significantly improve the efficiency of phosphorus and nitrogen uptake in maize plants.

Of the different treatments, the soil generated the plants with the highest plant mass in dry weight, but the thickness of the stem was lower than that obtained in the treatments with biosolids added at a concentration of 20% (*v*/*w*). According to what was reported by González-Flores et al. [82], biosolids favor the formation of thick stems and, consequently, the development of firmer plants. In the present study, fertilization with biosolids not only generated bending strength plants but also stimulated the development of cobs. The seedlings grown with chemical fertilizer were observed to be of good size and were apparently healthy; as in the other treatments, better root development was obtained in the inoculated treatment. However, the growth of the cob was not reached. 

## 4. Materials and Methods

### 4.1. Soil Collection

The soil was collected from an agricultural field situated in Ex-Hacienda Santa Inés, Nextlalpan, Edo, Mexico (19°41′32.4′′ N 99°04′39.4′′ W), by using the five-ring method described by Elizondo-Barrón et al. [83]. The weeds and residues from the previous crop were removed, and 200 g soil samples were taken 30 cm deep at five points. Then, the soil samples were mixed and stored for characterization (see Appendix A). 

### 4.2. Biosolids Collection

The biosolids, which are also known as sludge, were directly collected from a biological reactor at a wastewater treatment plant (WWTP) used for the food industry (see Appendix A). The collection followed the protocol for sampling in Title 40, Protection of Environment, part 503 [19]. The biosolids were collected in a 30 L polyethylene container under aseptic conditions during two different periods: summer and autumn 2021. The total volume was stored in a cold room at 4 °C until its later use. 

### 4.3. Soil and Biosolids Characterization

The characterization was carried out in relation to pH, humidity, electrical conductivity, organic matter content (OM), total solids (TS), total volatile solids concentration (TVS), total phosphorus and nitrogen, total coliforms, and heavy metal content. 

#### 4.3.1. pH

The pH of the soil and the biosolid was determined in triplicate from a sample/water suspension in a 1:2.5 ratio. The suspension was allowed to settle, and a sample of the supernatant was taken for measuring its pH with a potentiometer (Conductronic pH 120, Mexico City, Mexico) [84].

#### 4.3.2. Humidity

Humidity was determined by the gravimetric method proposed in NMX-AA-034-SCFI-2015 (Diario Oficial de la Federación N° 19, 11 February 2016). 7.5 g of soil (or biosolid) were weighed in a crucible of known weight and the water content was evaporated at 105 °C for 24 h. This test was conducted in quadruplicate, and the soil (or biosolids) humidity was calculated by weight difference.

#### 4.3.3. Soil Texture

The analysis was conducted in triplicate in 300 mL dilution bottles with 10 g dry soil samples, hydrated with 100 mL of distilled water. The hydration of the soil was conducted with vigorous agitation for 10 min, and later rest for 24 h. The sand, silt, and clay content were determined from the height data of the different soil textures observed in the dilution bottles [85]. The soil type was determined by interpolation on a triangular diagram of the basic textural classes of the soil (USDA, 1977).

#### 4.3.4. Organic Matter

The organic matter (OM, %) content in soil and biosolids samples was determined by ignition at 550 °C for 2 h. The analysis was conducted in triplicate in crucibles at constant weight (CWC, g) containing 5g samples of dehydrated soil (or biosolids) (DSM, g). The organic matter content was determined as follows [84]:(1)$$OM=\frac{\left(DSM-IDS\right)}{DSM}\times 100$$
where IDS (g) is the weight of the crucible with the ignited dry solids. 

#### 4.3.5. Electrical Conductivity

The electrical conductivity (EC) of the soil was determined in triplicate according to Zagal and Sadzawka [84]. Dehydrated soil samples of 20 g were moistened with 100 mL of deionized water. Wetting was performed with vigorous shaking for 30 min. The liquid phase was separated by filtration (11 μm Whatman) for conductivity analysis using a conductivity meter HQ14D (HACH, Ames, Iowa USA). The conductivity meter was calibrated with a standard solution 0.1 N of KCL (7.46 g of KCl calibrated to 1 L) of 12.9 dS m^−1^ at 25 °C.

#### 4.3.6. Total Solids

This analysis was conducted based on the APHA [86]. A 50 mL sample volume (SV, mL) of biosolids (0.3% *w*/*v*) was poured into a constant weight crucible (CWC, g) for moisture removal per triplicate at 105 °C. The crucible with the dry solids (CDS, g) was weighed on a Mettler H20 analytical balance (Toledo, Switzerland), and the mass of total solids (TS, g L^−1^) was calculated from the following equation:(2)$$TS=\frac{\left(CDS-CWC\right)}{SV}\times 1000$$

The analysis of total volatile solids (TVS, g L^−1^) in the biosolids was performed by the dry biosolid subsequent calcination at 550 °C for 20 min; the weight of calcined dry solids was identified as IDS (g). The TVS mass was calculated from the following [86]:(3)$$TVS=\frac{\left(CDS-IDS\right)}{SV}\times 1000$$

#### 4.3.7. Total Phosphorus and Total Nitrogen

Total phosphate content (mg kg^−1^) in soil (and biosolids) was determined by the molybdovanadate with acid persulfate digestion method (Method 10127) and the total nitrogen content in soil (and biosolids) by the persulfate digestion method (Method 10072), both described in the Hach manual [87]. The analysis was performed in a Multiparameter portable colorimeter DR900 (Ames, Iowa, USA) using the corresponding Hach colorimetric kits: 1.0 to 100.0 mg L^−1^ PO_4_^−3^ (HR) for the phosphate (# product:2767245) and the 2 to 150 mg L^−1^ (HR) for the nitrogen (# product: 2714100). Soil samples (or biosolids) of 30 g were suspended in 100 mL of distilled water. The insoluble phosphorus and nitrogen were determined after acid digestion (30 min) with 1N HCl of the soil (or biosolids) samples. The nitrate concentration (mg kg^−1^) was determined by the chromotropic acid method (Method 10020) proposed in the Hach manual (2014), using the 0.2 to 30.0 mg L^−1^ NO_3_^−^ (HR) colorimetric kit (#product:2605345).

Each analysis was carried out in triplicate in samples of the soil suspension diluted in the range (10^0^–10^−2^).

#### 4.3.8. Total Coliforms

The methodology proposed in NOM-113-SSA1-1994 [88] was applied. A serial dilution (1 × 10^−1^ to 1 × 10^−5^) of the biosolid was prepared from a 1 mL sample. Samples of 0.5 mL of each dilution were seeded in Petri dishes prepared with Violet Red Bile Agar (VRBA) for incubation at 30 °C for 72 h. The microbial count was determined in triplicate by the most probable number method and reported as colony-forming units per milliliter (CFU mL^−1^).

#### 4.3.9. Heavy Metals

The analysis of heavy metals was conducted by atomic absorption spectroscopy (GBC Scientific Equipment Ltd., Keysborough, Australia) based on NOM-004-SEMARNAT-2002 [20], which stipulates the maximum permissible limits of metals in the biosolid. The analysis was focused to the identification of the following metals: arsenic (Ar), cadmium (Cd), copper (Cu), chromium (Cr), lead (Pb), mercury (Hg), nickel (Ni), and zinc (Zn)). The analysis of each metal was performed separately [86].

### 4.4. Growth of Azotobacter nigricans in Soil Conditioned with Biosolids

*A. nigricans* was taken from the microbial collection of our research group [89]. The growth of *A. nigricans* in the biosolids and the agricultural soil selected for this study were tested with the biosolids at 25% (*v*/*v*); as a control, complete Rennie’s media (CR) was used. The study was conducted in three biological replicates in fifty-four Erlenmeyer flasks with a volume of 125 mL, with each having a sample volume of 25 mL. Each flask was inoculated with 1 mL of the *A. nigricans* strain (1 × 10^8^ CFU mL^−1^), and the propagation of this was carried out at 28 °C and 150 rpm for 120 h. Six flasks (three treatments and three controls) were taken every six hours during the first day and, subsequently, every 24 h for the analysis of microbial growth in terms of CFUs per milliliter (CFU mL^−1^).

### 4.5. Nitrogenase Activity in Azotobacter nigricans

An acetylene reduction assay (ARA) was performed as described by Postgate [90]. *A. nigricans* inoculated (*A. nigricans* 1 × 10^8^ CFU mL^−1^) in Rennie-modified medium (25 mL) was poured into 24 vials with a volume of 144 mL for the analysis in triplicate of the nitrogenase activity and microbial growth. The vials were sealed and incubated at 28 °C and 150 rpm for 72 h. Periodically (0, 3, 6, 9, 15, 24, 48, and 72 h), gas samples (0.1–0.5 mL) from three vials were taken and analyzed with gas chromatography by means of a VARIAN CP-3380 gas chromatograph (Spectralab Scientific, Markham, ON, Canada) that included an FID detector and a capillary column (0.25 mm). The detector and injector temperatures were set to 250 °C and 200 °C, respectively, and the column temperature was kept constant at 55 °C. Microbial growth was followed by the measurement of dry weight from the total microbial culture contained in the vials.

### 4.6. Indole Acetic Acid (IAA) Production by Azotobacter nigricans

*A. nigricans* was cultured in Rennie’s medium supplemented with tryptophan at 5 µg mL^−1^ or without tryptophan. The culture was carried out in triplicate with 18 Erlenmeyer flasks with a volume of 250 mL containing 50 mL of medium and 1 mL of inoculum (*A. nigricans* 1 × 10^8^ CFU mL^−1^). The flasks were incubated at 28 °C and 150 rpm for seven days. Six flasks, 3 from each treatment, were separated at 24, 48, and 120 h for the analysis of the indole acetic acid contained in the supernatant, which was separated by centrifugation at 10,000 rpm.

The supernatant was acidified with 1N HCl until a pH of 2.5 was reached, and it was extracted with ethyl acetate at a supernatant/solvent ratio of 1:2. The extraction process was carried out three times with equal volumes of solvent. The organic phase was recovered and evaporated at 40 °C. The IAA was then collected with 5 mL of methanol for subsequent analysis by using TLC Silica gel 60 F254 (Merck, Darmstadt, Germany) and liquid chromatography. Aliquots of 50 µL of the methanolic solution were placed on the bottom of a silica gel plate, and a solution sample of 98% IAA (No.: I3750-5G-A Sigma Aldrich, Steinheim, Germany) was used as a standard. The plates were then run with a solvent mixture of benzene–ethyl acetate–acetic acid (70/25/5, %/%/%), and spots were identified under ultraviolet light (254 nm).

IAA quantification via liquid chromatography was performed on cultures taken at 48 and 120 h, from which the extracted IAA was solubilized in 2 mL of methanol. The analysis was conducted by using a VARIAN 9002 HPLC-UV liquid chromatograph (Spectralab Scientific, Holland, Netherland) equipped with a Thermo Separation Product 3200 UV detector and a Gemini 5u C18 150 × 4.6 mm column (Phenomenex, Torrance, Los Angeles CA, USA). The sample volume injected was 50 µL. The concentration of the IAA produced was determined by using a calibration curve drawn with the IAA standard, which was included in the software for the same chromatograph. 

### 4.7. Solubilization of Tricalcium Phosphate by Azotobacter nigricans

In Rennie’s medium without nitrogen, tricalcium phosphate (TCP) was added at concentrations of 0.5, 1, and 1.5 g L^−1^, and dipotassium phosphate (4 g L^−1^) was used as a control. The medium (25 mL) was added to 84 Erlenmeyer flasks with a volume of 125 mL with 25 mL of each treatment and inoculated with 1 mL of *A. nigricans* (1 × 10^8^ CFU mL^−1^) for incubation at 28 °C and 150 rpm for 120 h; three biological replicates were made from this experiment. Periodically, three flasks of each treatment (a total of 12) were separated at 0, 8, 16, 24, 48, 72, and 96 h to analyze the soluble phosphorus, microbial growth, and pH of the culture medium. All media were prepared with deionized water to eliminate external sources of phosphorus.

### 4.8. Germination of Zea mays Seeds

Maize seeds (*Zea mays* L.) were obtained from the distributor ZARCO^®^. Disinfection was conducted by using a protocol described by García-Esquivel [89], which involved the use of 95% ethanol and a mixture of NaClO (0.5%) and Tween 80 (0.5%). First, the seeds were immersed in ethanol for 30 s and then rinsed with sterilized distilled water to remove any remaining ethanol. Next, the seeds were submerged in a NaClO (0.5%) and Tween 80 (0.5%) solution for 10 min, followed by thorough rinsing with sterilized distilled water to remove any remaining solution. To determine the germination percentage of the disinfected maize seeds, one hundred seeds were placed in Petri dishes prepared with water agar (4 g L^−1^). The germination occurred in a dark environment at room temperature, and the root length was recorded after 72 h. The seed germination criterium was a minimum root length of 5 mm, and the researchers discarded seeds with shorter roots according to the protocol established by Wang et al. [44].

### 4.9. Growth of Zea mays Seedlings in the Fähraeus System

To observe the development of roots, *Z. mays* seedlings with a mean radicle length of 10–15 mm were fixed on slides and immersed in a jar containing 30 mL of the treatments—soil, sterilized biosolids, 25% (*v*/*v*) B/S (sterilized biosolids in sterilized soil at 25%), and Long Ashton solution as a control [91]—with and without inoculum (*A. nigricans* 1 × 10^8^ CFU mL^−1^); the Fähraeus system was implemented in four biological replicates under sterile conditions. The seedlings’ growth was controlled at 25 °C with a photoperiod of 12 h and a light intensity of 1076.39 Lx. For a total of 8 treatments, 96 Fähraeus systems with one plant each were implemented. The radicular growth of the plants was recorded on days 9, 18, and 21, for which 32 flasks (4 of each treatment) were removed those days for the analysis of the number of roots, root length, and root thickness. The root morphology was also observed, and scanning electron microscopy (SEM) was used to observe of the association of *A. nigricans* with the plant roots. 

### Scanning Electron Microscopy Observation

The root segments were washed for 24 h in a 1% glutaraldehyde solution at room temperature, followed by three washes with 0.1 M Sorensen’s phosphate buffer at pH 7.2 and 4 °C for 15 min each. The samples were then post-fixed with 1% osmium tetroxide for 2 h and washed twice with water for 30 min each. Then, the roots were dehydrated via passage through increasing ethanol concentrations in water for 20 min each. The samples were dried to the critical point with CO_2_ in a samdri-780A (Tousimis, Rockville, MD, USA). The dried samples were affixed to a specimen stub with a conductive surface and coated with 30 nm gold for observations with a JSM-6390 scanning microscope (JEOL, Tokyo, Japan) at 10 kV.

### 4.10. Growth of Zea mays Plants in a Greenhouse

The *Z. mays* plants were grown in 15 L pots, each with 4.5 kg of soil. The soil was conditioned with biosolids in concentrations of 0, 15, and 20% (*v*/*w*) and commercial fertilizer, which was used as a positive control; soil with biosolids at 0%, was the negative control. Each of these treatments was implemented with and without *A. nigricans*. The total number of treatments was 8, with 10 repetitions each, one pot per repetition. The plants were grown in a greenhouse under the following conditions: temperature in the range of 14–28 °C, humidity in the range of 18–29%, photoperiod of 11 h, and watering every two days. The length of leaves and roots of maize plants, as well as their mass in units of dry weight per plant, were measured after 120 days of cultivation. The soil was analyzed in relation to its pH, texture, humidity, and organic matter content. 

A section of the corn plants’ leaves was cut and analyzed in triplicate for phosphorus and nitrogen content per gram of dry leaf. Digestion was performed as described in the protocol of Hu and Barker [92]. The phosphorus and nitrogen concentrations were determined with the corresponding Hach colorimetric methods.

### 4.11. Statistical Analysis

Results reported are the mean ± standard error (S.E) of three biological replicas. Statistical analysis was performed in PASW Statistics 18 V 18.0.0 (IBM Endicott, NY, USA). The Tukey test determined the significant differences between the means (*p* < 0.05).

## 5. Conclusions

The development of *Zea mays* plants due to the effects of their fertilization with biosolids collected from a food industry wastewater treatment plant and their inoculation with *Azotobacter nigricans*, a microorganism with the ability to solubilize phosphates and stimulate plant growth, was studied. According to the results obtained, biosolids favored the development of vigorous roots and the earliest emergence of the coleoptile in seedlings. *A. nigricans* promoted the development of an abundant radicular system in the plants, longer roots, and a more rapid growth of their stems and leaves. At the greenhouse level, the plants inoculated with *A. nigricans* showed an increase in the concentration of phosphates. The combination of the fertilization of *Z. mays* with the studied biosolids and *A. nigricans* favored the development of even longer roots compared to those generated in the systems implemented with *A. nigricans*, as well as the early emergence of cobs. The development of longer plants was expected; however, only the growth of more bending strength plants was observed. The best development of the plants inoculated with *A. nigricans* was attributed to the larger root surface of the plants, which was confirmed via scanning electron microscopy.

The better development of Z. *mays* plants fertilized with biosolids from the food industry and inoculated with *A. nigricans*, which is a plant-growth-promoting strain, is an indicator of the potential use of these two components as alternative sustainable fertilizers for application to crops that grow in semi-arid soils.

## Figures and Tables

**Figure 1 plants-12-03052-f001:**
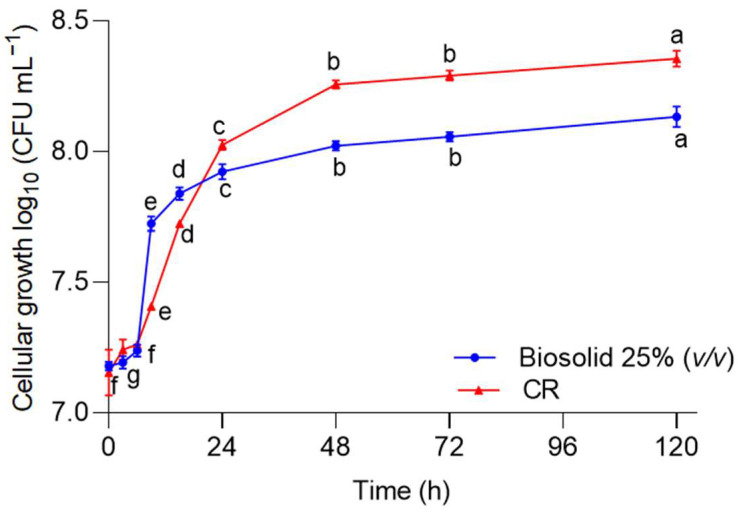
Growth kinetics of the *Azotobacter nigricans* culture in complete Rennie’s medium (CR) (▲) and biosolids at 25% (*v*/*v*) (●), both incubated at 28 °C and 150 rpm. The error bars represent the standard deviation of three independent determinations. Two-way ANOVA (*n* = 54). The normality requirement was met with the Kolmogorov–Smirnov test. Tukey’s test for α < 0.05: (F (time) = 1219.040, df = 8, *p* < 0.0001, χ^2^ = 0.996; F (media) = 30.122, df = 1, *p* < 0.0001, χ^2^ = 0.456; F(time,media) = 53.479, df = 8, *p* < 0.0001, χ^2^ = 0.922). Markers labeled with the same letter do not significantly differ.

**Figure 2 plants-12-03052-f002:**
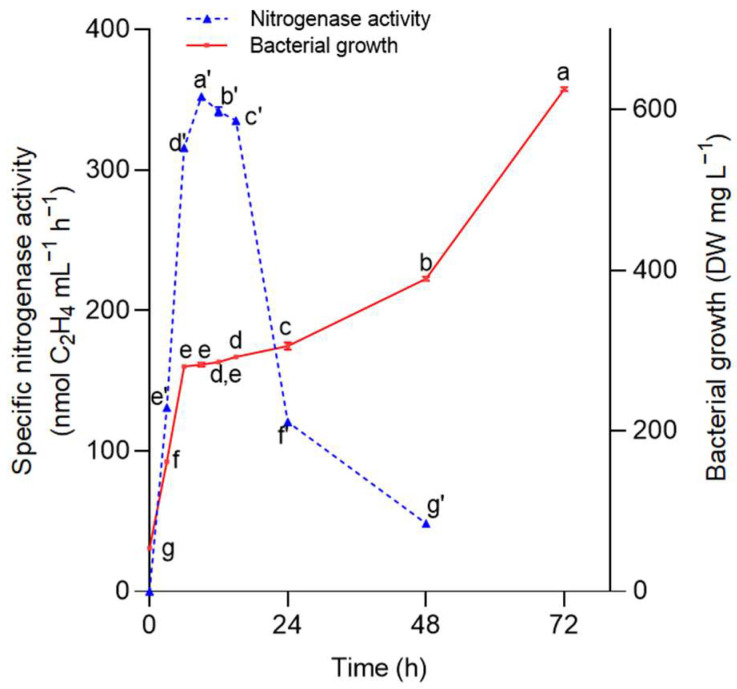
Culture at 28 °C and 150 rpm of *Azotobacter nigricans* in Rennie’s medium with sucrose as a carbon source. (■) Microbial growth in dry weight (DW); (-▲-) nitrogenase activity. The error bars represent the standard deviation of three independent determinations. One-way MANOVA (*n* = 27) (Tukey’s test for α < 0.05) (F (time)_bacterial growth_ = 72,952.813, df = 8, *p* < 0.0001, χ^2^ = 1; F(time)_nitrogenase activity_ = 69,984.806, df = 8, *p* < 0.0001, χ^2^ = 1). Markers labeled with the same letter do not significantly differ.

**Figure 3 plants-12-03052-f003:**
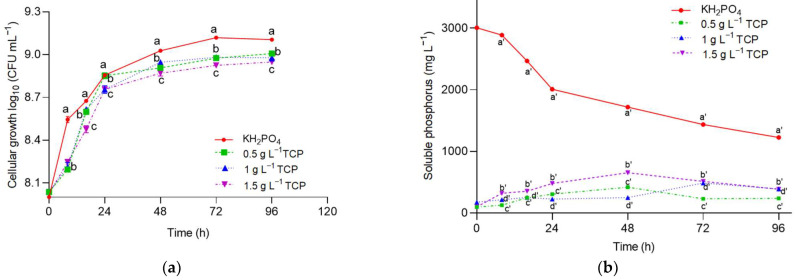
Solubilization of tricalcium phosphate (TCP) by *Azotobacter nigricans* with concentrations of 0.5, 1, and 1.5 gL^−1^ when incubated at 28 °C, 150 rpm, and with sucrose (5 gL^−1^) as a carbon source in Rennie’s medium. Control: soluble phosphorus (K_2_HPO_4_) at a concentration of 4 g L^−1^. (**a**) Microbial growth; (**b**) solubilized phosphorus. Two-way MANOVA (*n* = 56). (Wilks’Λ = 0, F(time) = 895.726; df_hypothesis_ = 12, df_error_ = 54; *p* < 0.0001; χ^2^ = 0.995. Wilks’Λ = 0, F(TCP concentration) = 4524.799; df_hypothesis_ = 6, df_error_ = 54; *p* < 0.0001, χ^2^ = 0.998; Wilks’Λ = 0, F(time, TCP concentration) = 821.808; df_hypothesis_ = 36, df_error_ = 54; *p* < 0.0001, χ^2^ = 0.998). Markers labeled with the same letter do not significantly differ (Tukey’s test for α = 0.05).

**Figure 4 plants-12-03052-f004:**
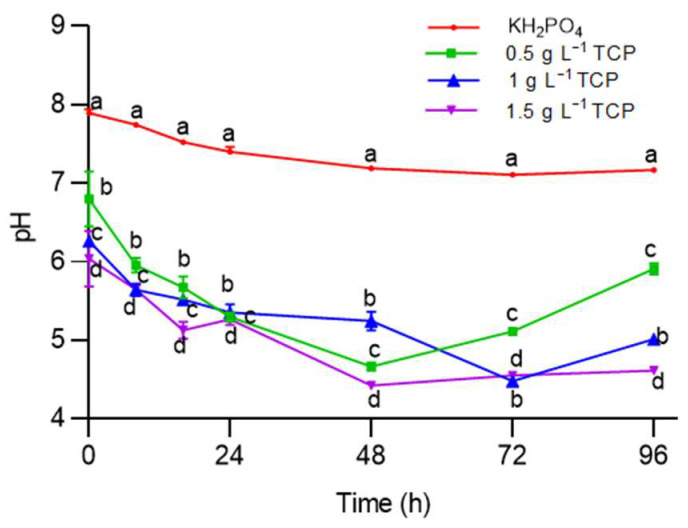
Variation in pH during the growth of *Azotobacter nigricans* in Rennie’s medium at 28 °C and 150 rpm. Two-way ANOVA (*n* = 56) (Tukey’s test for α < 0.05) was used with the nonparametric Wilcoxon test (F (time) = 113.176, df = 6, *p* < 0.0001; χ^2^ = 0.96; F (TCP concentration) = 804.281, df = 3, *p* < 0.0001, χ^2^ = 0.960; F (time, TCP conc.) = 8.073, df = 18, *p* < 0.0001, χ^2^ = 0.838). Markers labeled with the same letter do not significantly differ.

**Figure 5 plants-12-03052-f005:**
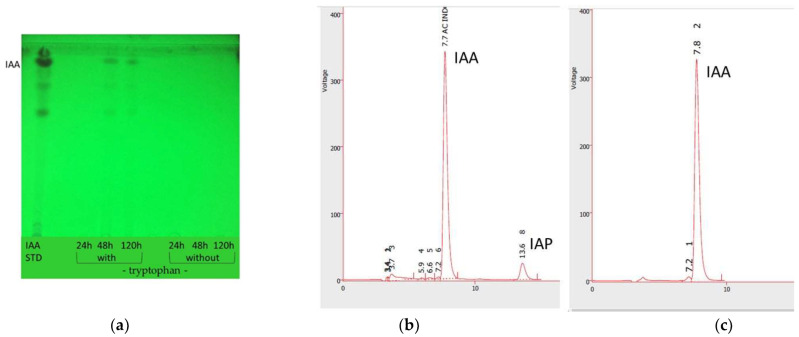
(**a**) Thin-layer chromatography (TLC) for verifying the production of indole acetic acid (IAA) during the growth of *Azotobacter nigricans* in Rennie’s medium prepared with (5 mg mL^−1^) and without tryptophan. The lanes on the left correspond to IAA and the medium with tryptophan sampled at 24 h, 48 h, and 120 h. The lanes on the right correspond to the medium without tryptophan sampled at 24 h, 48 h, and 120 h. (**b**) IAA chromatogram produced at 120 h in Rennie’s medium inoculated with *A. nigricans*. (**c**) Standard IAA chromatogram.

**Figure 6 plants-12-03052-f006:**
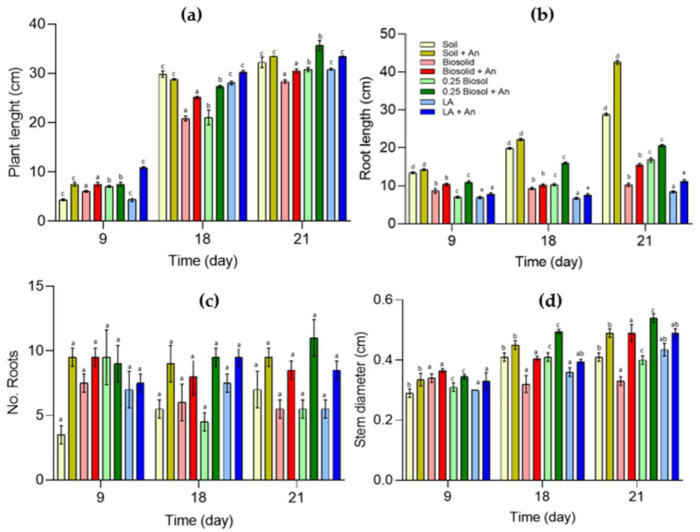
*Zea mays* seedlings grown with soil, biosolids at 25% (*v*/*v*), and Long Ashton with or without *Azotobacter nigricans* (An) (1 × 10^8^ CFU mL^−1^). Growth data in the Fähraeus system were recorded at 9, 18, and 21 days. (**a**) Seedling growth, (**b**) root growth, (**c**) number of roots, and (**d**) stem thickness. Three-way MANOVA (*n* = 120) (Appendix A). The comparison of means according to the Tukey test for α < 0.05 showed significant differences for three of the four variables studied: plant length, root length, and stem thickness; no significant differences were observed in the case of the number of roots. Bars labeled with the same letter do not significantly differ (Tukey’s test for α = 0.05).

**Figure 7 plants-12-03052-f007:**
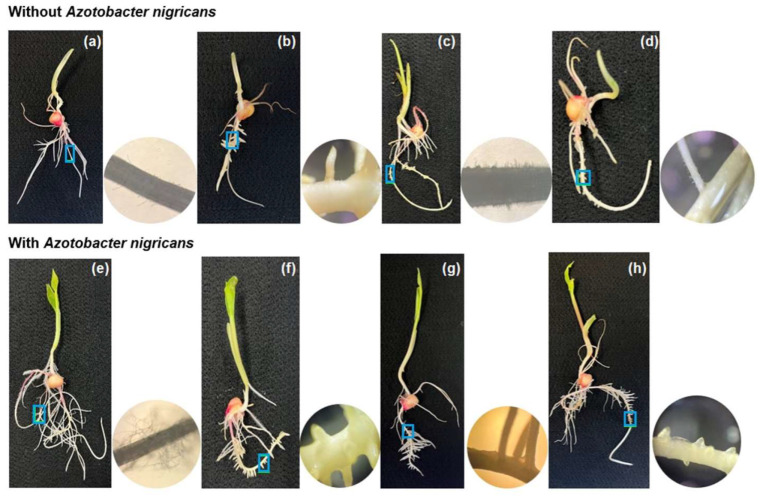
Seedlings and micrographs of the growth of *Zea mays* at 9 days. Treatments without and with *Azotobacter nigricans* (An) at a concentration of 1 × 10^8^ ± 6.4 × 10^6^ CFU mL^−1^: (**a**) soil, (**b**) biosolids, (**c**) soil with biosolids at 25% (*v*/*v*), (**d**) Long Ashton, (**e**) soil + An, (**f**) biosolids + An, (**g**) biosolids at 25% (*v*/*v*) + An, and (**h**) Long Ashton + An.

**Figure 8 plants-12-03052-f008:**
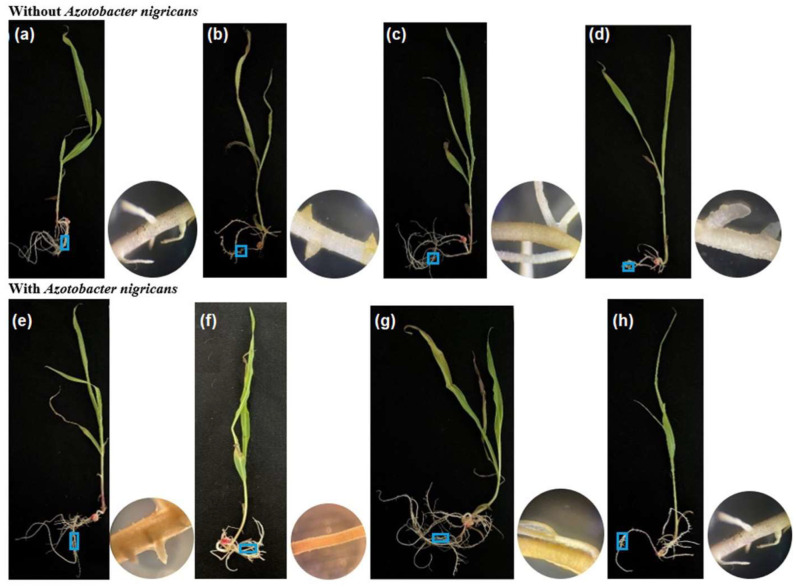
Seedlings and micrographs of the growth of *Zea mays* at 21 days. Treatments without and with *Azotobacter nigricans* (An) at a concentration of 1 × 10^8^ ± 6.4 × 10^6^ CFU mL^−1^: (**a**) soil, (**b**) biosolids, (**c**) soil with biosolids at 25% (*v*/*v*), (**d**) Long Ashton, (**e**) soil + An, (**f**) biosolid + An, (**g**) soil with biosolids at 25% (*v*/*v*) + An, and (**h**) Long Ashton + An.

**Figure 9 plants-12-03052-f009:**
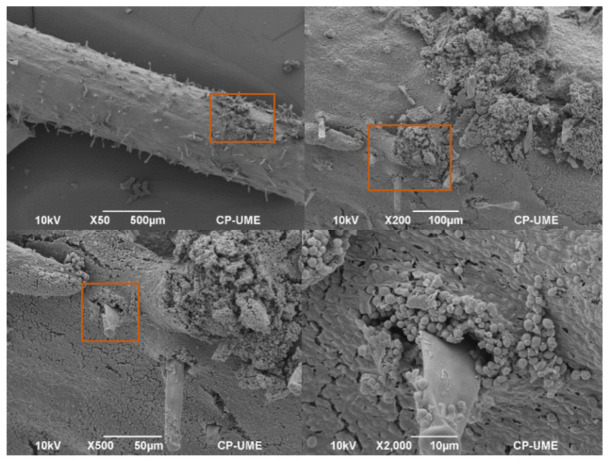
Colonization of *Azotobacter nigricans* in the roots of a *Zea mays* plant at 21 days of growth. Phosphorus source: dipotassium phosphate (K_2_HPO_4_), a soluble phosphorus.

**Figure 10 plants-12-03052-f010:**
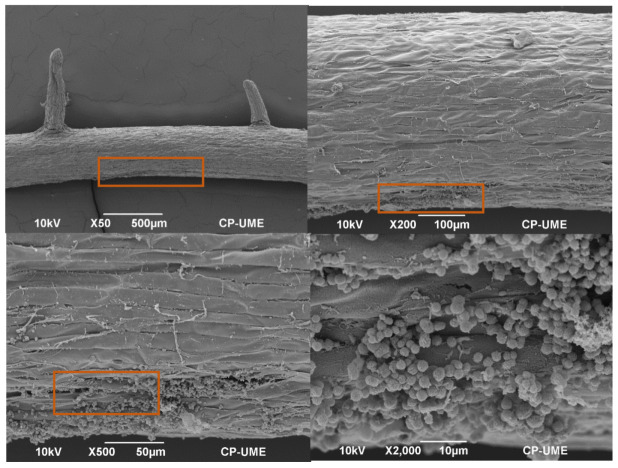
Colonization of *Azotobacter nigricans* in the roots of a *Zea mays* plant at 21 days of growth. Phosphorus source: tricalcium phosphate at 0.5 g L^−1^.

**Figure 11 plants-12-03052-f011:**
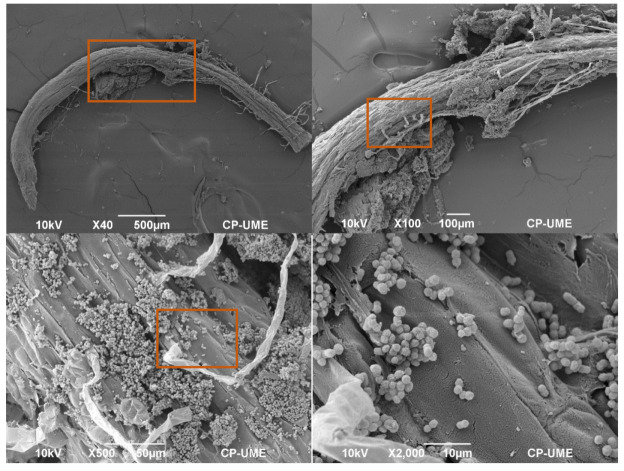
Colonization of *Azotobacter nigricans* in the roots of a *Zea mays* plant at 21 days of growth. Phosphorus source: biosolids at 25% (*v*/*v*).

**Figure 12 plants-12-03052-f012:**
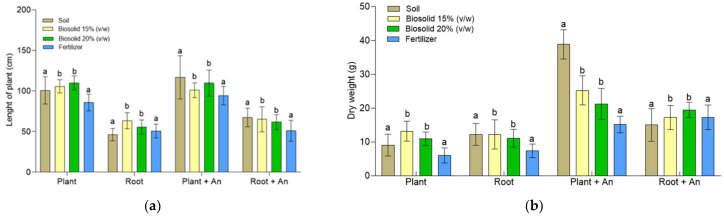
*Zea mays* plants grown in a greenhouse under controlled conditions with and without *Azotobacter nigricans* (An) inoculum. (**a**) Longitudinal growth and (**b**) plant growth as dry weight. Bars labeled with the same letter do not significantly differ (Tukey’s test for α = 0.05). Three-way ANOVA for the plant dry weight (*n* = 120). The normality test was fulfilled, but the Levene test was not satisfied. but the non parametric test for Kolmogorov–Smirnov is satisfied. The comparison of means according to the Tukey test for α < 0.05 showed significant differences for the inoculum (F(inoculum) = 50.824; df = 1; *p* < 0.0001, χ^2^ = 0.320), biosolids (F(biosolids) = 64.160; df = 2; *p* < 0.0001, χ^2^ = 0.543), and the inoculum–biosolid interaction (F(inoculum, biosolids) = 7.583; df = 2; *p* < 0.0001, χ^2^ = 0.123). Three-way ANOVA for leaf length (*n* = 120). The normality test and the Levene test were both satisfied. The comparison of means according to the Tukey test for α < 0.05 showed significant differences for the plant part (F(part plant) = 408.667; df = 1; *p* < 0.0001, χ^2^ = 0.791) and the biosolids (F(biosolids) = 20.390; df = 2; *p* < 0.0001, χ^2^ = 0.274).

**Figure 13 plants-12-03052-f013:**
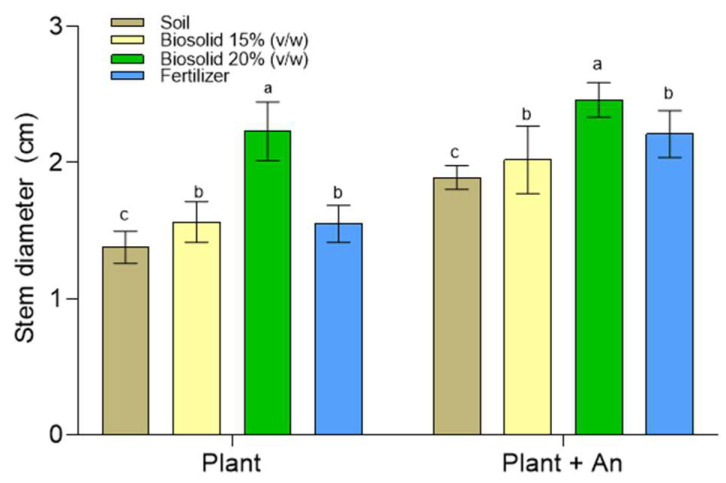
Stem diameter of *Zea mays* plants grown in a greenhouse for 120 days under controlled conditions with and without *Azotobacter nigricans* (An) as an inoculum. Two-way ANOVA (*n* = 80). The assumption of normality is not satisfied, but the non-parametric test of Kolmogorov–Smirnov is fulfilled. The comparison of means according to the Tukey test for α < 0.05 showed significant differences for the treatment (F = 69.006; df = 3; *p* < 0.0001; χ^2^ = 0.742), inoculum (F = 159.412; df = 1; *p* < 0.0001; χ^2^ = 0.689), and the treatment–inoculum interaction (F = 5.882; df = 3; *p* < 0.0001; χ^2^ = 0.197). Bars labeled with the same letter do not significantly differ (Tukey’s test for α = 0.05).

**Figure 14 plants-12-03052-f014:**
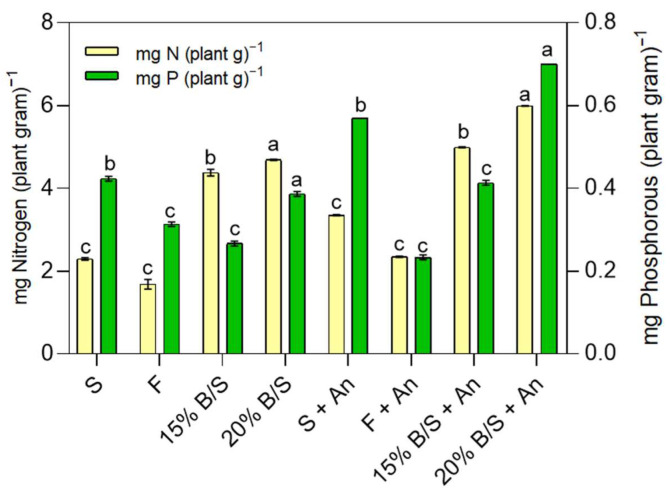
Content of nitrogen (N_2_) and phosphorus (P) by *Zea mays* plant leaves in greenhouse treatments implemented with biosolids at concentrations of 15 and 20% (*v*/*w*) (0.15 B/S and 0.20 B/S) and inoculated with *A. nigricans* (An). Positive control: fertilizer (F), negative control: soil (S). Two-way MANOVA (*n* = 16). Normality assumption is fulfilled according to the Shapiro–Wilk test. The non parametric test of Mann–Whitney is satisfied. The comparison of means according to the Tukey test for α < 0.05 showed significant differences for the treatment (Wilks’Λ = 0.000, F = 4486.296; df_hypothesis_ = 4, df_error_ = 22; *p* < 0.0001; χ^2^ = 0.999), inoculum (Wilks’Λ = 0.001, F = 6762.553; df_hypothesis_ = 2, df_error_ = 11; *p* < 0.0001; χ^2^ = 0.999), and the treatment–inoculum interaction (Wilks’Λ = 0.000, F = 179.702; df_hypothesis_ = 4, df_error_ = 22; *p* < 0.0001; χ^2^ = 0.970). Bars labeled with the same letter do not significantly differ. The comparison of means for this MANOVA was differentiated with bold letters.

**Table 1 plants-12-03052-t001:** Physicochemical and biochemical properties of the soil and biosolids under study.

Parameter	Agricultural Soil	Biosolids from the Food Industry	Ceiling Concentrations [19]
pH	8.10 ± 0.12	6.80 ± 0.03	
Electric conductivity (EC), dS m^−1^	1.05 ± 0.30	-	
Humidity (H), %	27.5 ± 0.1	-	
Organic matter (OM), %	8.70 ± 0.02	0.87 ± 0.20	
Cation exchange (CEC) cmol kg^−1^	32.00 ± 0.01	-	
Texture	sandy loam	-	
Clay, %	0.17	-	
Slime, %	28.74	-	
Sand, %	71.09	-	
Chemical parameters			
NO_3_^-^, mg kg^−1^	11.32 ± 0.10	320.10 ± 0.03	
Total nitrogen (N), mg kg^−1^	609.42 ± 0.10	1723.20 ± 0.03	
PO_4_^−3^, mg kg^−1^	4.44 ± 0.02	27.58 ± 0.60	
Total phosphorus (P), mg kg^−1^	13.62 ± 0.02	84.58 ± 0.60	
Metals			
As, mg kg^−1^	1.78 ± 0.12	1.12 ± 0.08	75
Cd, mg kg^−1^	<4 ± N.A.	4.19 ± 0.32	85
Cr, mg kg^−1^	<20 ± N.A.	<20 ± N.A.	-
Cu, mg kg^−1^	17.14 ± 0.91	<10 ± N.A.	4300
Pb, mg kg^−1^	23.61 ± 1.71	48.12 ± 3.49	840
Hg, mg kg^−1^	0.39 ± 0.04	1.98 ± 0.79	57
Ni, mg kg^−1^	<20 ± N.A.	<20 ± N.A.	420
Zn, mg kg^−1^	65.96 ± 3.12	34.74 ± 1.64	7500
Total solids (TSs), g L^−1^	-	15.90 ± 0.04	-
Total volatile solids (TVSs), g L^−1^	-	9.99 ± 0.50	-
Total organic carbon (TOC), mg L^−1^	-	631.00 ± 0.01	-
Chemical oxygen demand (COD), mg L^−1^	-	6050 ± 4	-
Total coliforms as colony-forming units (CFU), CFU mL^−1^	-	1.19 × 10^−5^ ± 0.01	-

N.A.: not available.

## Data Availability

Experimental data are included in the paper. Additional information will be provided on request.

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
