# Peer review of "Enhancing Phosphorus and Nitrogen Uptake in Maize Crops with Food Industry Biosolids and Azotobacter nigricans"

_plants, 2023, doi:10.3390/plants12173052_

Round 1
Reviewer 1 Report
The manuscript by Vera-Garcia et al. is on a topic of great interest, since lowering the widespread use of chemical fertilizers in agricultural settings has become an important goal in sustainable, low input agriculture. The manuscript explores the use of biosolids from waste treatment plants and Azotobacter nigricans as fertilizers of maize crops. The manuscript generally reads well, but fixing some writing errors would improve its quality. The authors describe a series of logical experiments, but the manuscripts would be improved by considering the following recommendations. I was also not able to download the supplemental data; the link provided in the manuscript did not work, making it impossible for me to evaluate the data.
- The introduction is too brief, and fails to emphasize what knowledge or practical gap is this work going to fill. Azotobacter has been widely used in agriculture for the properties mentioned by the authors. The use of biosolids from waste treatment plants is nothing new. It is also surprising that there is no mention about arbuscular mycorrhizal fungi which play important roles in acquiring phosphate for host plants. Overall, the introduction is insufficiently developed.
- The soil appears to be from a single location, with nearly neutral pH, favorable for both plants and Azotobacter – which is not found in more acidic soils. This is insufficient to reflect most field conditions.
- I don’t understand why was Azotobacter nigricans use in this study collected from polluted soil of a particular location. The authors do not describe validation tests done to ensure that what they collected from the polluted soil is A. nigricans, not another strain of Azotobacter, and is not a mix including other microbes collected from the respective location.
- The biosolid material is expected to contain lots of microorganisms; coliforms are tested, but a more thorough analysis would identify many more types of microbes. So, the SEM analysis to observe the association of maize with Azotobacter is questionable. How do the authors know that what they see in SEM images is Azotobacter and not some other bacteria from the biosolids?
- I suppose that, by ‘experimental unit’, the authors mean each condition: soil, soil with biosolid, etc. (paragraph 4.5). What is not mentioned is the sample size for each condition; how many plants were treated in each experimental unit? Furthermore, there is no mention of biological replicates. Three biological replicates from three independent experiments is the norm to generate meaningful data.
- It is not clear why the authors measured Azotobacter nitrogenase activity in the Renie-modified medium, instead of using the growth substrates used in the experiments described in this work. The relevance of nitrogenase activity is within the context of the growth substrate in which maize plants grow, not in a number of vials inoculated with Azotobacter.
- I also don’t understand why the authors measured IAA production by A. nigricans, disregarding the other hormones this bacterium is known to produce: cytokinins and gibberellins. Once again, not clear why the measurement was done in inoculated media in vials, rather that the A. nigricans-supplemented growth substrates. Other studies demonstrated that A. nigricans fix nitrogen as free-living bacteria, produce various hormones, and solubilize phosphate, so these measurements from test tubes are not relevant here. Readers would be interested to see how A. nigricans promotes plant growth together with the biosolids, and measure these parameters within the context of the proposed growth substrates.
- Fig. 5 needs to be updated. Labeling on the TLC plate is too small. There are about three bands in the IAA sample and the media; which one is IAA? HPLC results should also be shown as chromatograms, indicating the IAA peak and how was its concentration calculated. It is surprising that the total methanolic extract applied on the TLC plate shows the exact same bands as the IAA standard. A methanolic extract should be more complex and more bands should be visible on plate, especially since UV light was used to visualize, rather than some specific staining. It is important to add a second panel with the HPLC chromatogram of the same sample to show all the peaks detected by HPLC and clarify which one is IAA.
- The micrograph insets in Figures 7 and 8 are not well explained, specifically which exact segments they represent. I suggest marking the exact area on the corn plant that is magnified in each inset.
- Line 296: magnification is not a measurement in micrometers. Please fix this error.
- Please provide the data with the biometric analysis of seedlings under all experimental conditions. Percentages are included in text, but this is not clear. For example, in lines 264-265, it is stated that the “coleoptile is short, between 8-17% compared to that observed in the other treatments”. This is vague; please show the data. The same goes for coleoptile and root thickness.
The manuscript generally reads well, but fixing some writing errors would improve its quality.
Author Response
Answers to the comment made by the Reviewer 1. Manuscript ID: plants-2440271
The authors appreciate the suggestions and comments made by the reviewers. They allowed us to substantially improve the presentation of the proposed manuscript for its possible publication in the Plants Journal.
The specific questions were answered individually. The corrected manuscript was submitted for review of English grammar.
Answers in attached file

Reviewer 2 Report
The MS titled “Enhancing phosphorus and nitrogen uptake in Maize crops with food industry biosolids and Azotobacter nigricans” has scientific relevance and quality to be published in Plants Journal.
Generally, the work is coherent and it is easy to understand. The MS title fully reflects the content of the paper. Initially, the article makes a good impression and is wonderfully illustrated with nice photos of root formation (with and without bacteria) and also SEM micrographs; the results are new and quite interesting. However, I cannot recommend MS in presented form for publication because it contains a lot of errors and inaccuracies.
First of all, the information presented in Table 1 looks incorrect and raises many questions. It is not clear why the total nitrogen content is significantly less than the nitrate (NO3-) content. What the authors mean by “total nitrogen” and why is it shown in the form of molecular nitrogen (N2)? The same applies to the content of phosphates and total phosphorus, the concentration of which is also, for some reason, less than phosphates. It is not clear where the authors got the “Recommended values for agricultural soils”. The Ref. 34 they provided is not correct (the article 34 does not contain this data). The indicated values themselves are also questionable and doubtful, especially for nitrates - 1549.45 mg/kg. This value is more than the order of magnitude higher than the recommended values in many countries: for example, in Russia, the maximum allowable concentration of nitrates is 130 mg/kg. The very wording “Recommended values” with regard to heavy metals looks extremely incorrect, since it would be appropriate to use “Maximum permissible concentrations”. It is not clear what references to them are indicated – CLEA, SGV (no decryption anywhere)? The recommended (or allowable) content of heavy metals is a mistake (can you seriously recommend 80 mg/kg of Pb?). The abbreviations – DQO, UFC are incorrect presented. In general, the presentation of data in the article, the design of tables, figures, captions to them is very careless, contains many inaccuracies, incorrectness, and mistakes. There is no clarity in the presentation of the Ðœaterials and Methods subsection, there is no uniformity in the presentation of data on the introduction of biosolids amount (somewhere the percentage is indicated, in other cases - the proportion; how can there be the concentration range from 0 to 100% (line 592)? Actually, according to the figures, 3 concentrations were used – 15, 20 and 25%? Somewhere the B/S ratio is indicated by weight (w/w), somewhere by volume (v/v), etc.
The Abstract needs to be corrected, there are many inaccuracies and ambiguities, as well as Keywords that are not very correct (especially the “development of plants”, “plant inoculant interaction” – better “plant-bacterial (or -microbial interaction”) and do not fully reflect the essence of the work. English needs to be improved. In the MS there are a lot of grammatical errors and typos. In many cases, the authors write abbreviations first (without decoding), and the decoding itself is shown either at the end of the article, or it is not. Many of the abbreviations are both correct and incorrect (“CFU” – “UFC”).
Therefore, I encourage authors to make all necessary corrections and resubmit the MS “Enhancing phosphorus and nitrogen uptake in Maize crops with food industry biosolids and Azotobacter nigricans” in Plants Journal for further review again.

English needs to be improved. In the MS there are a lot of grammatical errors and typos.
Author Response
Answers to the comment made by the Reviewer 2. Manuscript ID: plants-2440271
The authors appreciate the suggestions and comments made by the reviewers. They allowed us to substantially improve the presentation of the proposed manuscript for its possible publication in the Plants Journal.
The specific questions were answered individually. The corrected manuscript was submitted for review of English grammar.
Answers in attached file

Reviewer 3 Report
The manuscript reports on the effects of biosolids with and without inoculation of Azotobacter on the germination and plant growth of Zea mays. This is a very interesting study where the microscope images stand out.
The introduction is a bit poor. There is need for a better explanation of what biosolids are and what is the rational behind the study to have chosen the ones used in the current study.
The objectives are merely descriptive and there are no scientific hypothesis to be tested. Hypotheses need to be incorporate to the manuscript and, please, make sure that you do not write predictions but proper scientific hypotheses.
In the material and methods section there are some aspects that need to be considered and improved.
On line 608 it is said that ‘The seed germination criterium was a minimum root length of 5 cm’. This must be a typo as 5 cm is far too long for a root, so germination should be presumed at the protrusion of the radicle as it is considered in most of the scientific literature having to do with both seed germination studies and agronomic ones.
What are the units used to measure plant development (lines 616-619).
SEM: give it in full the first time it is used (line 618).
Statisical analyses: you say that ANOVA was used to test for what? This needs to be expanded upon. In addition, did you test that data met the requirements of normality and homocedasticity for an ANOVA to be conducted? The only ‘n’ value given in the study is 100 for seed germination. As germination only goes between 0 and 1, ANOVA cannot be applied. For the rest of the analyses, what are the samples sizes and the number of replicates used. You need to work this harder to convince me that you are using the correct statistical analyses.
The best part of the results section is a constant mix between results and discussion. In consequence, this section needs to be completely rewritten attaching to facts with no interpretation, that has to be preserved for the discussion section. Results are very long also due to the extensive writing and extremely long sentences that need to be simplified.
The use of patterns on the columns on the figures make it very hard to see the differences. Please, use solid colors for the bars and make use the online facility where colors can be seen.
The discussion is well written and properly documented. However, there are aspects presented in the results section, like germination, that have not been dealt with.
The conclusion presented are not such but mere summaries of the results. Conclusions are to be written after the scientific hypotheses have been proven or rejected. As the study does not present hypotheses to be tested consequently there are not conclusions. This section has to be written ex-novo and in view of the scientific hypotheses to be proposed as well.
For more specific comments see below and the annotated manuscript.

Author Response
Answers to the comment made by the Reviewer 3. Manuscript ID: plants-2440271
The authors appreciate the suggestions and comments made by the reviewers. They allowed us to substantially improve the presentation of the proposed manuscript for its possible publication in the Plants Journal.
The specific questions were answered individually. The updated manuscript was submitted for edition of English grammar.
Answers in attached file

Round 2
Reviewer 1 Report
This manuscript version is improved, and I appreciate the authors response to most of my previous comments.
Author’s answer to Question 5 is not satisfactory. While numbers of plants are mentioned, there is no indication that they conducted independent biological replicates as separate experiments carried out from start to finish at different times, with new/different Azotobacter cultures. Using a large number of plants in the same experiment does not substitute true biological replicates. Section 4.10 (Statistical analysis) – lines 763-765 is the only place where the authors mention replicas/replicates, without defining what replica means within the context of their work.
The authors’ response to Question 6 is not satisfactory. Thus, the question remains open/unanswered. Nitrogenase activity measured in the Rennie’s medium is not relevant to the maize growth in the presence of biosolids and Azotobacter. If the authors wanted to establish a link between Azotobacter's ability to fix nitrogen in vials and in the presence of biosolids and maize plants, then same measurements should be done under both experimental conditions.
The response to Question 7 is also unsatisfactory. Azotobacter’s IAA production in vials with Rennie’s medium is not relevant to it’s possible IAA production in the presence of the biosolid while growing maize plants.
There are still English writing errors in this updated manuscript version. For example, the last sentence in the Abstract: “The better development of the plants was attributed to the increase in their root surface of the plant.” This should be reformulated.
Line 71: “Collaborations” is not the right word to use between inanimate materials (biosolids) and microorganisms.
Author Response
The answers were uploaded in a pdf file

Reviewer 2 Report
It can be seen that the authors have done a great job and made a lot of changes and amendments. However, I cannot recommend the MS “Enhancing phosphorus and nitrogen uptake in Maize crops with food industry biosolids and Azotobacter nigricans” for publication in its present form, because in the version 2 that I downloaded from the site and tried to read, this is completely impossible to do. The authors have made their new corrections without deleting the old version of the text in many sentences, which causes misunderstanding and requires serious editing. Many phrases both in the abstract and in the text of the MS are worded incorrectly. English needs editing. The design of the article needs to be improved according to the requirements of the journal.
The responses to previous comments are also not entirely satisfactory. It raises a big question and doubt how the authors reacted a little "frivolously" to the remark about the data in Table 1, simply rearranging them in some cases or changing the value by an order of magnitude, is it possible to be sure that these are real numbers (there was not even a justification why did it happen). Moreover, in the MS 2 version, which is provided on the site, some of the old numbers have not been removed, and therefore it would not be clear at all what was meant.
I encourage the authors to add a more detailed description of methods for analyzing soil and biosolids characteristics, rather than just citing sources (Table 2). In response to the question, the authors incorrectly wrote the author's surname “Kendall” instead of Kjeldahl, you should be more careful!
In Abstract the sentence: “At the greenhouse level, the Z. mays plants fertilized with biosolids at concentrations of 15 and 20% (v/w) and A. nigricans favored the development of more robust plants (5% thicker) …” is essentially not formulated correctly, it is not clear what "robust plants" means, and what does “5% thicker” have to do with it? It is not clear "thicker" what? In addition, 5% is a very small value, usually within the standard error, and relying on such minor changes, in my opinion, is not very correct. This sentence should be revised.
I recommend that authors make corrections and attach two files - one "clean" already fully corrected and the second in edit mode, so that reviewers can fully appreciate all the editing.

English has been corrected, but still needs to be improved. Perhaps if the authors send the correct version of the article, it will be better seen.
Author Response
The answers were uploaded in a pdf file

Reviewer 3 Report
The authors have done a very good job revising the manuscript and it meets now the requirement for publication in the Jounal Plants. Thanks for the effort made to meet these scientific standars.
I have not comments.
Author Response
The answers were uploaded in a pdf file

Round 3
Reviewer 1 Report
The authors improved the quality of the manuscript, but important questions that have not been addressed prevent me from recommending this version of the manuscript for publication. Here are a few points I made earlier that have not been addressed.
Question 4: An alternative method to SEM should have been used to validate that what is seen by SEM is ONLY Azotobacter. The simple statement that the media was sterile is not scientific, measurable evidence.
Question 5: I appreciate the details the authors provide, in particular about the sample size, but this is for one biological replicate. There is no indication that they actually conducted independent biological replicates.
The answer to Question 6 is only a justification as to why they conducted the experiment that way they described, and does not address my point. Making measurements in the Renie-modified medium is not relevant to how Azotobacter may affect maize growth, therefore the authors did not address this very important point.
The answer to Question 7 is also unsatisfactory for the same reasons I explained above.
It is improved compared to the previous version.
Author Response
The authors appreciate the reviewer's comments. We agree that there could be few samples analyzed in the Fähraeus system (section 4.9). The experiment will be repeated in order to have a 3x3 experimental design. From this same experiment we could identify the microorganism that we saw by SEM on the roots of the seedlings. This analysis could be by colony and cellular morphology. Regarding questions 6 and 7. The authors accept the reviewer's suggestion, however we will not know if the nitrogenase activity is caused by the assimilation of nitrogen from the air or by the nitrogen contained in the solid matrices. Analysis of nitrogenase activity and indole acetic acid (IAA) production will be implemented, but in a system that does not include Zea mays seedlings. The analytical method for analyzing nitrogenase activity is not compatible with the requirements for plant growth. In the case of the IAA analysis, it could not be identified since the plant consumes this metabolite.

Reviewer 2 Report
The authors have done a lot of work to correct MS titled “Enhancing phosphorus and nitrogen uptake in Maize crops with food industry biosolids and Azotobacter nigricans”, but not enough to recommend it for publication in this form. There are still a lot of comments, to a greater extent they are all related to the negligence of the article in accordance with the rules of the journal. I don't want to list them all, I marked them in proof pdf. But I'm not sure that I could list them all, so I recommend the authors to pay special attention to some points. First of all, this concerns the correct citation of literature in the article. Since, according to the rules of the journal, citations are in square brackets in the order of appearance of references, the year after the author's surname is usually not indicated. In my opinion, subtitle 2.2 is poorly titled, it is not clear what the authors meant by writing only the species name (Azotobacter nigricans)? It should be clarified. At the same time subtitle 2.3.2. too long with extra unnecessary details. Subtitles 4.1. and 4.2. also need correction (it is better to clarify, for example, “Soil collection or sampling”, etc.). And in general, all the subtitles are designed incorrectly, I strongly recommend the authors to pay attention to the correctness of their design. As for the legends in the Figures, it is not clear why the authors write in some cases all the words in capital letters, and in others with small ones (for example in Figure 2 “Specific nitrogenase activity” and “Bacterial Growth”, etc.)? The data in Figure 14 are presented extremely incorrectly: if the authors calculated the content of nitrogen and phosphorus in leaves by dry weight (this should also be clarified in the methodology, otherwise it simply indicates that the calculation was made “per gram of leaves”), then incorrect numbers are indicated (possibly an error when calculated, the values are incorrect by at least an order of magnitude). Perhaps the authors did the recalculation for the whole plant, but then this needs to be clarified in the methodology and the legend corrected (should be “mg N (P) plant–1”). The side legends also do not look correct, as does the caption for Figure 14 since the authors did not measure the assimilation of nitrogen and phosphorus, but only their content!
I can recommend this article for publication in the Plants journal after making all the corrections and taking into account the comments.

There are typos and inaccuracies that need to be corrected.
Author Response
Please see the response in the attached document
